# Research

behaviour/ecology/evolution

*Drosophila*, protein restriction, selection, life-history traits, trans-generational effect

**Author for correspondence:**
Pankaj Yadav
e-mail: ypankaj1451981@gmail.com

# Trans-generational effect of protein restricted diet on adult body and wing size of *Drosophila melanogaster*

## Sudhakar Krittika and Pankaj Yadav

Fly Laboratory no. 210, Anusandhan Kendra-II, School of Chemical & Biotechnology, SASTRA Deemed to be University, Thanjavur 613401, Tamil Nadu, India

(iD) SK, 0000-0003-0733-5375; PY, 0000-0003-2246-305X

Protein restriction (PR) has established feasible trade-offs in *Drosophila melanogaster* to understand lifespan or ageing in a nutritionally challenged environment. However, the phenotypes of body size, weight and wing length respond according to factors such as flies' genotype, environmental exposure and parental diet, and hence their understanding is essential. Here, we demonstrate the effect of long-term PR diet on body size, weight, normal and dry wing length of flies subjected to PR50 and PR70 (50% and 70% protein content present in control food, respectively) for 20 generations from the pre-adult stage. We found that PR-fed flies have lower body weight, relative water content (in males), unaltered (PR50%) and higher (PR70%) relative fat content in males, smaller normal and dry body size when compared with control and generations 1 and 2. Interestingly, the wing size and pupal size of PR flies are smaller and showed significant effects on diet and generation. Thus, these traits are sex and generation dependent along with a diet interaction, which is capable of modulating these results variably. Taken together, the trans-generational effect of PR on fitness and fitness-related traits might be helpful to understand the underpinning mechanisms of evolution and ageing in fruit flies *D. melanogaster*.

# 1. Introduction

Organisms vary in body size not only across species but also within a particular species. The variations in the body composition can influence phenotypic traits like body size, body weight, etc., while these trait variations can be attributed to various environmental and genetic factors [1–3]. The environmental factors that can influence organismal body size and weight, including wing length (especially in insects) can be nutrition [4], temperature [5,6], crowding [4,7], latitudinal clines [8] and

certain cases of laboratory selection pressures for faster development [9], etc. Body size, weight and wing length are certain parameters that ensure the overall fitness of organisms including fruit flies. Thus, variations in these phenotypes can be used to understand the genotypic changes that are bound to occur [10].

Fruit flies *Drosophila melanogaster* for the past three decades, have been widely used as a model organism for studying ageing via nutritional approaches including diet restriction (DR), food dilution, intermittent feeding, etc. [11–13]. The diet of fruit flies commonly comprises carbohydrates and proteins as the major source, with lipids, vitamins, minerals present in minor quantities. Restricting protein source (yeast) in the fly food is a type of dietary implementation (protein restriction; PR henceforth) and is seen to influence a range of fitness and fitness-related traits such as lifespan, fecundity, stress resistance, activity, development time, etc. [13–16]. Interestingly, PR is known to influence traits like body size, weight and wing length, wherein variations in yeast concentration can significantly alter the body size and weight of the flies, and also have an influence on their wing length in a single generation itself (environment effect; [1,2,17]). This might be owing to the sudden change in the protein composition; while it is also necessary to assay long-term (genetic effect) effect of PR. A high protein diet can yield unaffected pupal size [18], while the long duration of high protein increased body mass [19], also decreased body weight and fat levels [20].

PR from the pre-adult stage is highly debatable as some studies suggest its negative effect on lifespan, body size, fecundity [1,21,22]; while studies also claim its positive effect on lifespan and other traits [13,23,24]. The adult body size need not necessarily influence the lifespan of the organism raised under varied nutritional conditions [25]. Because nutrition during the pre-adult stage largely determines the size of the adult upon eclosion [1,2,25] alongside the influences of juvenile hormones [26], it is essential to study the long-term effect of PR and not just one or two generations [27]. Moreover, assaying for one or two generations might address the immediate effect of parents or grandparents' diet on the offspring and whether it is maternally inherited or not, remains unanswered [28,29]. The current study assays a trans-generational effect of 20 generations (gen 1, 2 and 20) and does not deal with understanding the mode of inheritance (maternal or paternal).

Here, we address the effect of 50% and 70% yeast concentrations (as against the control ad libitum (AL) diet) across generations 1, 2 and 20. The PR concentrations of 50% and 70% which have been used are based on the preliminary PR studies on the lifespan of the flies (S. Krittika, P. Yadav 2022, unpublished data). This study will address the effect of PR on a single generation, its offspring (generation 2), and also the long-term effect (generation 20) of the corresponding protein restriction. The assessed traits include body size, weight, relative water and fat content, pupal size, and wing length in the normal and dry conditions of the fly body. After 20 generations of PR implementation, the PR males and females have lower body weight when compared with their control in their respective generations and within the PR generations. Interestingly, the relative water content is higher in females and not in males despite the long-term PR diet. Because the body weight of the PR flies is lower after generations, we found it necessary to assess the relative fat content of the flies. We found that the PR70% males have higher fat storage after 20 generations, while females showed no difference. Moreover, the flies at generation 20 have lower body size when compared with gen 1 and 2 and their control, showing that the body size and weight might be positively correlated. Because the body size is an adult trait, we measured the intermediate pupal size. The PR50 flies at gen 1 and 2 showed the highest pupal size when compared with PR70 and control, while their size was similar at the end of 20 generations, which might be reflected as a part of smaller adult body size as well. Lastly, because wing length itself can be an indicator of body size [30], measuring the same revealed that PR had a diet and generation-dependent effect on producing flies with shorter wings. Thus, after 20 generations, PR diets produce flies with smaller bodies and wings, lower weight, unaltered pupal size and relative water content (in females). Thus, this study benefits the understanding of the influence of diet and/or the genetic effect (generational study) in mediating variations in the assayed traits.

# 2. Material and methods

## 2.1. Fly culture and maintenance

The control and PR imposed flies are maintained on 21-day discrete generation cycles with egg collection done exactly on the twenty-first day of the previous generation cycle. The control flies are fed with AL food, while the PR stocks are fed with 50% and 70% yeast when compared with the control (PR50% and

PR70%; henceforth). The egg collection for control and PR stocks are done in their respective AL and PR diets (AL diet but with 50% and 70% yeast for PR50 and PR70 stocks, respectively). The flies upon eclosion are transferred to plexi-glass cages ($25 \times 20 \times 15$ cm) and are supplemented with their corresponding food. For the following experiments, approximately 30–40 eggs per vial were collected for control and PR and maintained at a temperature of approximately 25°C (±0.5°C), humidity of approximately 70%, and light intensity of approximately 250 lux in 12 L : 12 D cycles. The diet manipulations were done only in the yeast concentration present in the control food, wherein we used instant dry yeast from Gloripan.

## 2.2. Normal body weight and dry weight

We measured the normal and dry body weight of freshly eclosed flies collected every 2 h intervals. The eggs were collected from the PR stocks over a 2 h window and kept under a 12 L : 12 D cycle. Post eclosion, the virgin male and female flies were separated by anesthetizing with $CO_2$. For weighing the normal body weight, flies were weighed post anesthetization using ether (to maintain the flies in the anesthetized state for a longer duration), after which the flies were discarded. For the dry body weight assay, the virgin males and females were killed by freezing and were dried for 36 h at 70°C as per the protocol followed elsewhere [9]. The normal and dry body weight assay was assessed by weighing a group of 10 males or 10 females per vial, and five such vials of randomly chosen flies from the control and the DR stocks were weighed. The body weight of flies was measured using a weighing balance from UniBloc (Shimadzu) AUX220. The relative water content of the flies was calculated by dividing the water content (normal body weight-dry weight) by the normal body weight of the flies as reported elsewhere [8]. The relative fat content was assessed by dividing the fat content (dry weight-fat free dry weight) by the dry weight of the flies [8].

## 2.3. Body size/length and wing length

The protocol for egg collection until the separation of virgin male and female flies for this assay is similar to that followed for the body weight assay. The flies' body size and wing length were measured under a microscope, wherein 30 virgin males and females from the control and DR stocks were assayed. The body size and the wing length of the anesthetized males and females were measured using a microscope from Olympus with a normal ruler (with least count 0.5 mm).

# 3. Results

## 3.1. Normal and dry body weight

To check whether the body size and weight are proportional to each other under the imposed PR, we assayed body weight and size (normal and dry) of the AL and PR stocks at gen 1, 2 and 20 of diet imposition. ANOVA followed by post hoc multiple comparisons using Tukey's test on the normal body weight of the freshly eclosed male and female fruit flies showed a statistically significant effect of diet (D; $F_{2,72} = 193.04$, $p < 0.0001$), generation (G; $F_{2,72} = 99.63$, $p < 0.0001$), sex (S; $F_{1,72} = 490.70$, $p < 0.0001$) and their interaction (D × G × S; $F_{4,72} = 9.54$, $p < 0.0001$; table 1). The results show that PR50 (males) and PR70 (males and females) have significantly lower body weight at gen 20 when compared with their previous generations (figure 1a); while PR50 females at gen 20 have lower body weight than gen 2, but not gen 1. The PR50 and PR70 (males and females) have lower body weights at all tested generations against their respective control. Thus, with PR and long-term restrictions, the adult body weight is lower. To assess whether this lower body weight is owing to the water content, we weighed the dry weight of the flies.

   As expected, the ANOVA on dry body weight of flies revealed a statistically significant effect of D ($F_{2,72} = 106.75$, $p < 0.0001$), G ($F_{2,72} = 37.78$, $p < 0.0001$), S ($F_{1,72} = 289.74$, $p < 0.0001$) and their interaction (D × G × S; $F_{4,72} = 10.8$, $p < 0.0001$; table 1). *Post hoc* multiple comparisons by Tukey's test revealed significantly decreased dry body weight of PR50 (males and females) at gen 20 and PR50 flies have lower dry weight when compared with their gen 1, but not gen 2. Interestingly, PR70 males show no effect of generations as their dry weight is not different, while PR70 females at gen 20 have lower weight when compared with gen 1 and 2 (figure 1b). The effect of diet shows that at gen 1 and 2, the PR50 females are lower in weight, while the others weigh similar to AL. However, surprisingly at gen

**Table 1.** ANOVA details of the normal and dry body weight and relative water and fat content of long-term PR imposed flies.

| assay | effect | d.f. | MS effect | d.f. error | MS error | F | p< |
|---|---|---|---|---|---|---|---|
| normal body weight | diet (D) | 2 | 114.29 | 72 | 0.59 | 193.04 | 0.0001 |
| | gen (G) | 2 | 58.99 | 72 | 0.59 | 99.63 | 0.0001 |
| | sex (S) | 1 | 290.52 | 72 | 0.59 | 490.7 | 0.0001 |
| | diet × gen (D × G) | 4 | 8.65 | 72 | 0.59 | 14.61 | 0.0001 |
| | diet × sex (D × S) | 2 | 0.88 | 72 | 0.59 | 1.48 | 0.2342 |
| | gen × sex (G × S) | 2 | 0.34 | 72 | 0.59 | 0.57 | 0.5654 |
| | diet × gen × sex (D × G × S) | 4 | 5.65 | 72 | 0.59 | 9.54 | 0.0001 |
| dry body weight | diet (D) | 2 | 8.7003 | 72 | 0.0815 | 106.75 | 0.0001 |
| | gen (G) | 2 | 3.079 | 72 | 0.0815 | 37.78 | 0.0001 |
| | sex (S) | 1 | 23.61 | 72 | 0.0815 | 289.74 | 0.0001 |
| | diet × gen (D × G) | 4 | 1.015 | 72 | 0.0815 | 12.46 | 0.0001 |
| | diet × sex (D × S) | 2 | 2.607 | 72 | 0.0815 | 31.99 | 0.0001 |
| | gen × sex (G × S) | 2 | 0.13 | 72 | 0.0815 | 1.61 | 0.2064 |
| | diet × gen × sex (D × G × S) | 4 | 0.88 | 72 | 0.0815 | 10.8 | 0.0001 |
| relative water content | diet (d) | 2 | 0.0077 | 72 | 0.0007 | 11.4 | 0.0001 |
| | gen (G) | 2 | 0.002 | 72 | 0.0007 | 3.03 | 0.0547 |
| | sex (S) | 1 | 0.0026 | 72 | 0.0007 | 3.87 | 0.0531 |
| | diet × gen (D × G) | 4 | 0.0070 | 72 | 0.0007 | 10.38 | 0.0001 |
| | diet × sex (D × S) | 2 | 0.0225 | 72 | 0.0007 | 33.3 | 0.0001 |
| | gen × sex (G × S) | 2 | 0.0035 | 72 | 0.0007 | 5.12 | 0.0083 |
| | diet × gen × sex (D × G × S) | 4 | 0.0057 | 72 | 0.0007 | 8.46 | 0.0001 |
| relative fat content | diet (d) | 2 | 0.0026 | 72 | 0.0163 | 0.16 | 0.8506 |
| | gen (G) | 2 | 0.0792 | 72 | 0.0163 | 4.87 | 0.0104 |
| | sex (S) | 1 | 0.0011 | 72 | 0.0163 | 0.07 | 0.7922 |
| | diet × gen (D × G) | 4 | 0.1587 | 72 | 0.0163 | 9.76 | 0.0001 |
| | diet × sex (D × S) | 2 | 0.0508 | 72 | 0.0163 | 3.12 | 0.0501 |
| | gen × sex (G × S) | 2 | 0.0329 | 72 | 0.0163 | 2.02 | 0.1399 |
| | diet × gen × sex (D × G × S) | 4 | 0.2327 | 72 | 0.0163 | 14.3 | 0.0001 |

20, PR50 and PR70 (males and females) dry weight difference is prominent and weighes lower than the control. Thus, the results confirm that the body weight of PR flies is lower than the AL owing to long-term diet implementation.

Because there exists a difference between the normal and dry body weights across generations, we assessed the relative water content in the PR flies. ANOVA on the relative water content revealed a statistically significant effect of D ($F_{2,72} = 11.4$, $p < 0.0001$) and its interaction with generation (D × G; $F_{4,72} = 10.38$, $p < 0.0001$ and D × G × S; $F_{4,72} = 8.46$, $p < 0.0001$; table 1), but not G ($F_{2,72} = 3.03$, $p < 0.0547$) and S ($F_{1,72} = 3.87$, $p < 0.0531$). The relative water content of the PR50 flies at gen 2 is comparatively higher than gen 1, while PR70 males and females have higher water content at gen 1 and 20, respectively (figure 1c). For the effect of diet across generations, at gen 1, the PR50 and PR70 flies are similar to that of AL, while at gen 2, the PR50 males have similar water content to that of its respective control, while PR70 males and PR50 females exhibit lower and higher water content,

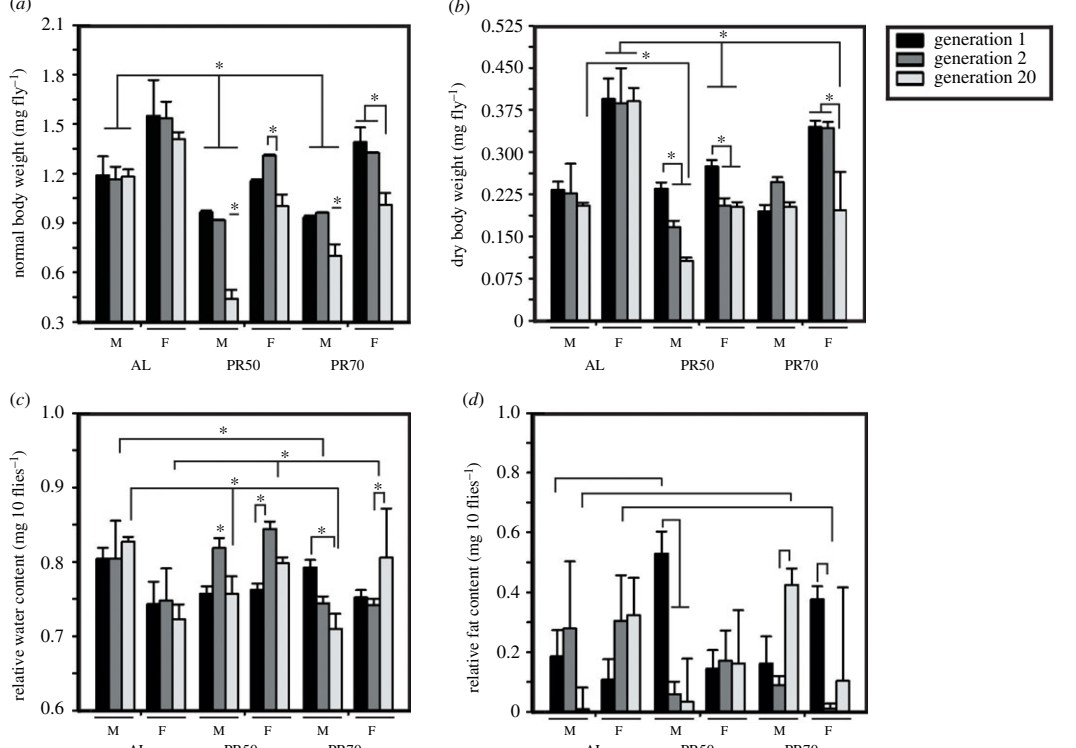

**Figure 1.** Low weighed males and females under the PR diet for 20 generations. The normal (*a*) and dry (*b*) bodyweight of the flies (varying across generations), shows PR flies weighing lower than that of AL flies at the end of gen 20. The effect of diet on the relative water content (*c*) is prominent, wherein after 20 generations of PR diets, male and female flies possessed lower and higher water content when compared with AL, respectively. The graph represents diet on the *x*-axis and body weight (*a,b*), relative water content (*c*), relative fat content (*d*) in the *y*-axis. The bars and error bars are represented as the mean ± standard deviation (s.d.). The asterisks on the bars indicate significance levels wherein the *p*-value is less than 0.05. G1, G2 and G20 represent gen 1, 2 and 20, respectively, while M and F represent males and females, respectively.

respectively. However, interestingly at gen 20, PR males have lower relative water content, while in PR females' it is higher. Thus, long-term PR has facilitated higher water content in PR females and not males.

Assessing the direct fat content in flies upon PR can give us information on the fat metabolism in flies. ANOVA on the relative fat content showed a statistically significant effect of G ($F_{2,72} = 4.87$, $p < 0.0104$) and its interaction (D × G; $F_{4,72} = 9.76$, $p < 0.0001$ and D × G × S; $F_{4,72} = 14.3$, $p < 0.0001$; table 1), but not D ($F_{2,72} = 0.16$, $p < 0.8506$) and S ($F_{1,72} = 0.07$, $p < 0.7922$). The post hoc multiple comparisons by Tukey's test revealed significantly higher fat content in PR70 males at gen 20, while for females, it was not significant (figure 1*d*). Interestingly, at gen 2, PR70 females stored lesser fat than the AL females. PR50 flies did not show any difference in their fat content except for PR50 males which showed higher fat than AL in gen 1 (figure 1*d*). Therefore, with respect to the fat content, long-term PR has facilitated higher fat content in PR70 males alone after 20 generations, while no significant change was observed in females. This reiterates that the fat accumulation might be sex-dependent and diet dependent as well, given that trans-generational effect is put into consideration.

## 3.2. Normal and dry body size

The flies maintained on PR50% and 70% for 20 generations from the pre-adult stage were measured for their normal and dry body size. ANOVA on the normal body size of freshly eclosed adult males and females showed a statistically significant effect of diet (D; $F_{2,522} = 38.51$, $p < 0.0001$), generation (G; $F_{2,522} = 98.19$, $p < 0.0001$), sex (S; $F_{1,522} = 611.60$, $p < 0.0001$) and their interaction (D × G; $F_{4,522} = 21.79$, $p < 0.0001$ and D × S; $F_{2,522} = 17.52$, $p < 0.0001$) (table 2 and figure 2*a*), but not D × G × S ($F_{4,522} = 2.07$, $p < 0.0830$). Furthermore, post hoc multiple comparisons using Tukey's HSD test revealed the generation effect is prominent in the PR flies as their body size at gen 20 is comparatively smaller than the previously tested generations (1 and 2). The interaction between all three tested factors is not

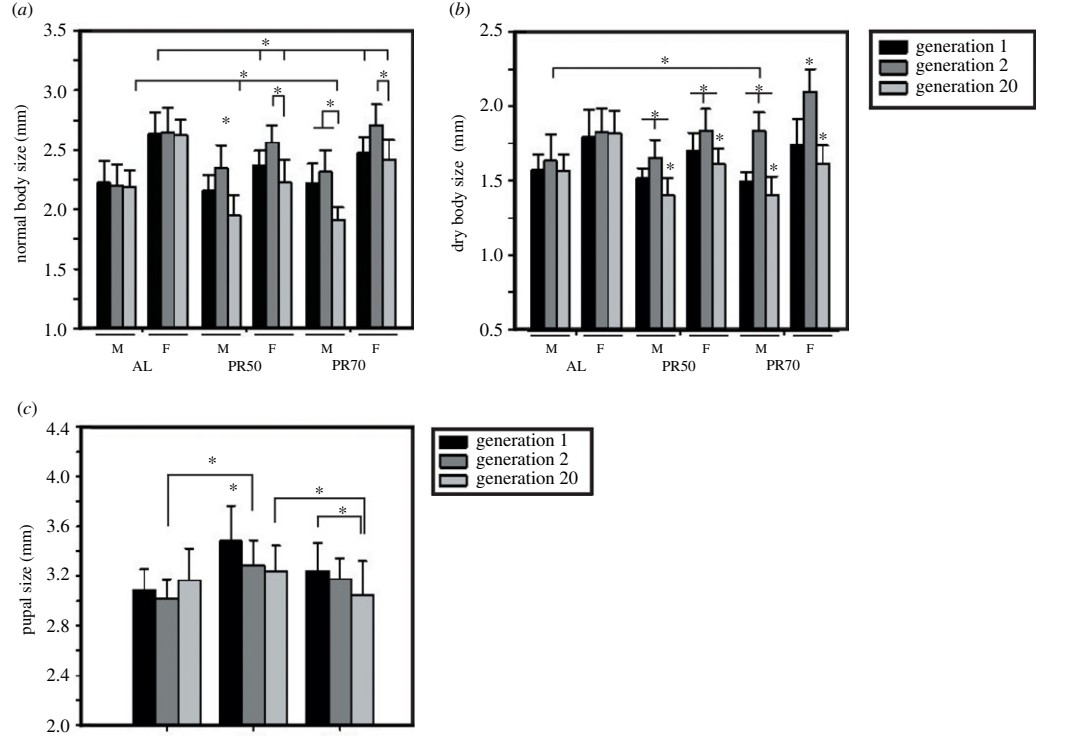

**Figure 2.** Smaller flies and unaltered pupal size owing to the PR diet for 20 generations. The effect of diet and generation on the normal (*a*) and dry (*b*) body size of the flies are variable, wherein the normal body size of PR flies is lower than their control at the end of 20 generations. The pupal size (*c*) of PR50 flies was the highest at gen 1 when compared with PR70 and control, but after 20 generations were similar to AL. The graph represents diet on the *x*-axis and body size (*a*,*b*), pupal size (*c*) on the *y*-axis. All other details are the same as in figure 1.

**Table 2.** ANOVA details of the normal and dry body size of long-term PR imposed flies.

| assay | effect | d.f. | MS effect | d.f. error | MS error | F | p< |
|---|---|---|---|---|---|---|---|
| normal body size | diet (D) | 2 | 1.036 | 522 | 0.027 | 38.51 | 0.0001 |
| | gen (G) | 2 | 2.641 | 522 | 0.027 | 98.19 | 0.0001 |
| | sex (S) | 1 | 16.45 | 522 | 0.027 | 611.60 | 0.0001 |
| | diet × gen (D × G) | 4 | 0.586 | 522 | 0.027 | 21.79 | 0.0001 |
| | diet × sex (D × S) | 2 | 0.471 | 522 | 0.027 | 17.52 | 0.0001 |
| | gen × sex (G × S) | 2 | 0.143 | 522 | 0.027 | 5.32 | 0.0052 |
| | diet × gen × sex (D × G × S) | 4 | 0.056 | 522 | 0.027 | 2.07 | 0.083 |
| dry body size | diet (D) | 2 | 0.374 | 522 | 0.018 | 21.23 | 0.0001 |
| | gen (G) | 2 | 2.870 | 522 | 0.018 | 163.13 | 0.0001 |
| | sex (S) | 1 | 6.394 | 522 | 0.018 | 363.37 | 0.0001 |
| | diet × gen (D × G) | 4 | 0.724 | 522 | 0.018 | 41.12 | 0.0001 |
| | diet × sex (D × S) | 2 | 0.025 | 522 | 0.018 | 1.42 | 0.2438 |
| | gen × sex (G × S) | 2 | 0.0004 | 522 | 0.018 | 0.03 | 0.9729 |
| | diet × gen × sex (D × G × S) | 4 | 0.015 | 522 | 0.018 | 0.86 | 0.4907 |

significant probably because body size is capable of higher perturbation to environmental factors. The effect of diet shows that at gen 1, PR males do not show any difference in body size, while PR females are smaller. However, surprisingly at gen 2, the PR-fed males are larger than AL males, while

**Table 3.** ANOVA details of the pupal size of long-term PR imposed flies.

| assay | effect | d.f. | MS effect | d.f. error | MS error | F | p< |
|---|---|---|---|---|---|---|---|
| pupal size | diet (D) | 2 | 1.452 | 261 | 0.049 | 29.34 | 0.0001 |
| | gen (G) | 2 | 0.394 | 261 | 0.049 | 7.96 | 0.0004 |
| | diet × gen (D × G) | 4 | 0.292 | 261 | 0.049 | 5.91 | 0.0001 |

females are similar in size as that of AL females; but at gen 20, PR males and females are smaller than the AL flies. Thus, after 20 generations, the PR produces smaller flies when compared with AL even though minor fluctuations in their body size were observed at gen 1 and 2.

Further, ANOVA followed by multiple comparisons by Tukey's HSD test on the dry body size showed a statistically significant effect of D ($F_{2,522} = 21.23$, $p < 0.0001$), G ($F_{2,522} = 163.13$, $p < 0.0001$), S ($F_{1,522} = 363.37$, $p < 0.0001$) and their interaction (D × G; $F_{4,522} = 41.12$, $p < 0.0001$; table 2 and figure 2b), but not D × S ($F_{2,522} = 1.42$, $p < 0.2438$) and D × G × S ($F_{4,522} = 0.86$, $p < 0.4907$). All the PR flies at gen 2 are bigger when compared with gen 1 and 20, except for PR70 males whose dry body size is similar to that observed at gen 1. The effect of diet on the dry body size revealed that PR flies are similar in size to AL at gen 1, while at gen 2 the PR70 (males and females) are bigger than AL. Similar to the results of normal body size, the PR flies are smaller than their control flies at gen 20 and might be owing to the factors discussed earlier with normal body size. Surprisingly, there exist changes in the response of PR diet on the normal and dry body size, showing that the normal body size and dry body size might not be equivalent and the difference between them is not constant, and that reason might be attributed to the various forms of storage reserves.

## 3.3. Pupal size

ANOVA followed by Tukey's HSD test on the pupal size showed a statistically significant effect of D ($F_{2,261} = 29.34$, $p < 0.0001$), G ($F_{2,261} = 7.96$, $p < 0.0004$) and their interaction (D × G; $F_{4,261} = 5.91$, $p < 0.0001$; figure 2c and table 3). *Post hoc* multiple comparisons by Tukey's test showed that among PR50 flies across generations, gen 1 was the highest, while at gen 2 and 20 they were similar. Across diets, PR50 flies had a higher pupal size in gen 1, 2 and gen 20 when compared with AL and PR70 flies, thus, showing that the PR50 flies have a higher pupal size when compared with the control in all the generations, but within its generations, the observed highest pupal size at gen 1 might have been a startle response for PR.

## 3.4. Normal and dry wing length

Post body size assessments, we intended to assay the wing length as it is commonly thought to be a measure of body size as mentioned earlier. ANOVA on the normal wing length showed a statistically significant effect of D ($F_{2,522} = 72.28$, $p < 0.0001$), G ($F_{2,522} = 28.07$, $p < 0.0001$), S ($F_{1,522} = 301.92$, $p < 0.0001$) and their interactions (D × G; $F_{4,522} = 10.6$, $p < 0.0001$; and D × G × S; $F_{4,522} = 4.6$, $p < 0.0012$; table 4). Interestingly, multiple comparisons by Tukey's test within diets across generations shows that gen 20 females have wing length similar to gen 2, while the males have smaller wing length compared to gen 2 (figure 3a). Moreover, PR50 females have smaller wing lengths when compared with AL in all tested generations; while PR70 females have smaller wings when compared with AL in the first generation alone (figure 3a). The effect of diet shows that PR flies (males and females) have shorter wings than AL flies at gen 20. Thus, the concept of wing length as a measure for body size might not hold in the presence of dietary parameters influencing them across generations.

ANOVA on dry wing length of fruit flies revealed a significant effect of D ($F_{2,522} = 13.46$, $p < 0.0001$), G ($F_{2,522} = 33.84$, $p < 0.0001$), S ($F_{1,522} = 112.05$, $p < 0.0001$) and their interactions (D × G; $F_{4,522} = 26.19$, $p < 0.0001$ and D × G × S; $F_{4,522} = 5.33$, $p < 0.0003$; table 4). Multiple comparisons of dry wing length by Tukey's test revealed results similar to the normal wing length wherein PR flies at gen 20 had significantly smaller wings than the control (figure 3b). Thus, even though PR yields flies with shorter wings at the end of 20 generations (similar to dry body size), it is not true across generations, while it explicitly shows significant interaction between diet, sex and generations.

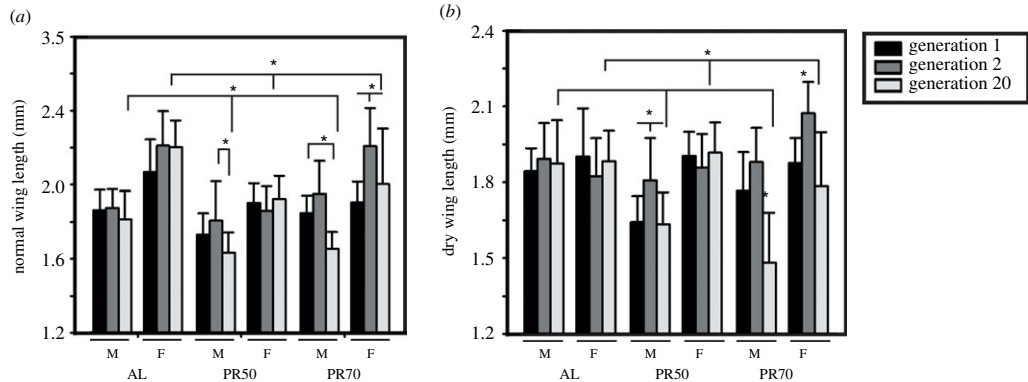

**Figure 3.** Flies' wing length reduced owing to long-term PR diet imposition. The normal (*a*) and dry wing length (*b*) of the PR flies is smaller than the AL flies post 20 generations, but the pattern of variations across generations is different from that witnessed for body size. The graph represents diet on the *x*-axis and wing length on the *y*-axis. All other details are the same as in figure 1.

**Table 4.** ANOVA details of the normal and dry wing length of long-term PR imposed flies.

| assay | effect | d.f. | MS effect | d.f. error | MS error | F | p< |
|---|---|---|---|---|---|---|---|
| normal wing length | diet (D) | 2 | 1.757 | 522 | 0.024 | 72.28 | 0.0001 |
| | gen (G) | 2 | 0.682 | 522 | 0.024 | 28.07 | 0.0001 |
| | sex (S) | 1 | 7.338 | 522 | 0.024 | 301.92 | 0.0001 |
| | diet × gen (D × G) | 4 | 0.258 | 522 | 0.024 | 10.6 | 0.0001 |
| | diet × sex (D × S) | 2 | 0.223 | 522 | 0.024 | 9.19 | 0.0001 |
| | gen × sex (G × S) | 2 | 0.453 | 522 | 0.024 | 18.62 | 0.0001 |
| | diet × gen × sex (D × G × S) | 4 | 0.112 | 522 | 0.024 | 4.6 | 0.0012 |
| dry wing length | diet (D) | 2 | 0.287 | 522 | 0.021 | 13.46 | 0.0001 |
| | gen (G) | 2 | 0.723 | 522 | 0.021 | 33.84 | 0.0001 |
| | sex (S) | 1 | 2.393 | 522 | 0.021 | 112.05 | 0.0001 |
| | diet × gen (D × G) | 4 | 0.559 | 522 | 0.021 | 26.19 | 0.0001 |
| | diet × sex (D × S) | 2 | 0.608 | 522 | 0.021 | 28.49 | 0.0001 |
| | gen × sex (G × S) | 2 | 0.224 | 522 | 0.021 | 10.51 | 0.0001 |
| | diet × gen × sex (D × G × S) | 4 | 0.114 | 522 | 0.021 | 5.33 | 0.0003 |

# 4. Discussion

## 4.1. Normal and dry body weight

Our results are similar to the results of another study [28] that shows the effect of diet (parental) on different traits (including body weight) of fruit flies, and it also suggests that these observed differences might be owing to the maternal effects and the long-term DR imposition. In the PR50 and 70% flies, there might be effect of parental diet on the normal body weight of the flies as suggested elsewhere [28,29], because the PR flies showed lower body weight at the end of 20 generations. Moreover, the lower body weight of PR flies after 20 generations can be thought to be in line with the study of Kristensen *et al.* [19], which reported the protein-rich diet for 17 generations yielded bigger and thereby fatter flies when compared with the control. But surprisingly, PR70 males have higher fat content than AL at the end of 20 generations, which makes it contrary to Kristensen *et al.* [19], even though the study reported the implementation of protein-rich diet. A difference exists in the fat and water content between the males under PR50 and PR70, which could probably be owing to the

difference in protein levels in the fly food. It can also suggest that adaptation to reduced protein levels might be quicker and since our data is one among the very few studies testing different concentrations of long-term PR diets, further in-depth studies are needed. Further, we did not expect variations in the AL body weight across generations, and convincingly their body weight and relative water content were unaffected, thereby providing convincing results for the control flies. Interestingly at gen 20, the dry weight of PR70 females is lower than that of AL, while its water content is higher and the fat content is similar to AL, thereby showing that the weight was probably contributed by the water content. However, in the case of males, they were similar to that of AL and their water and fat content are inversely proportional to each other. The difference in relative fat content upon sex-based effect is similar to that reported elsewhere [19]. Thus, long-term PR implementation suggests the existence of a plastic response to the diet when compared with the genetic effect in the case of dry weight and sex [19].

## 4.2. Body size and pupal size

The body size of PR flies is smaller at gen 20 when compared with their previous generations and control. These results are contrary to the study of Chippindale *et al*. [31], which reported that a bigger adult body size is associated with increased fitness of the flies. Moreover, the fitness of the organism is assessed based on its reproductive capacity and the ability to withstand stress, our results might have a positive effect despite a smaller body size. Because the males of *D. melanogaster* prefer smaller females for the first mating and then undergo adaptive discrimination [32] or plasticity for mate selection by males [33], we can conclude that body size may be one of the many traits that are assessed to choose a potential mate but not a primary one. Hence, the smaller body size of the PR flies might not be a threat to their mate choice in our study, even though the fecundity of our flies remains to be tested.

The pupal size of PR50 flies recorded the highest size when compared with the control and PR70 flies, while gen 1 flies of PR50 yielded the highest pupal size compared to gen 20 (figure 2*c*). Because a high protein diet did not confer any change in the pupal size of the flies, but a high carbohydrate diet resulted in smaller pupa [18], it is surprising to see pupal size difference upon protein restriction. This is in line with the results of Deas *et al*., which suggests more susceptibility of pupal mass change in poor diet than that of rich nutrient diet, in addition to exhibiting effects of parental and grandparental diet [34]. Overall, our results also show that diet and generation have a differential role of different traits as suggested elsewhere [34].

## 4.3. Normal and dry wing length

The concentrations of nutrients (yeast and sugar) in the fly diet play a more important role in the wing length of females than in males [17]. In our study, because the concentration of sugar was kept constant, the observed variations in wing length show that the yeast alone can modulate this trait. This is contrary to Güler *et al*. [17], where female's wing length varies with yeast manipulations while males vary with sugar level variations. The PR flies showed smaller wings after 20 generations depicting the generational effect while there exist variable results owing to sex difference. There are various other factors capable of modulating wing length like temperature and latitudinal clines [3,35], wherein care was taken to avoid such temperature perturbations. There exists a difference in the PR and generation effect on the trend of body size and wing length variations and probably, is in contrast to the study reported elsewhere [30], which stated that wing length can serve as a parameter for estimating a fly's body size.

Data accessibility. The primary data used to generate graphs can be accessed at https://doi.org/10.5061/dryad.8cz8w9gqz [36].

Authors' contributions. S.K.: conceptualization, data curation, formal analysis, investigation, methodology, validation, visualization, writing—original draft, writing–review and editing; P.Y.: conceptualization, data curation, funding acquisition, investigation, methodology, project administration, resources, software, supervision, visualization, writing—original draft, writing—review and editing. All authors gave final approval for publication and agreed to be held accountable for the work performed therein.

Competing interests. The authors declare that they have no conflict of interest.

Funding. S.K. acknowledges the Department of Science and Technology–Government of India, for the INSPIRE fellowship (IF170750). P.Y. acknowledges the Science and Engineering Research Board (file no. CRG/2019/003184 and YSS/2015/000354), Department of Science and Technology–Government of India, India for the financial support and SASTRA Deemed to be University, Thanjavur (TN), India for the infrastructure.

Acknowledgements. We thank Mr N. Ramesh, Mr Muralitharan and Mr Jayaraj (SASTRA University) for helping us with population maintenance.

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
