## [Peer Review File · Royal Society Open Science]

Review History

RSOS-211325.R0 (Original submission)

Review form: Reviewer 1

Is the manuscript scientifically sound in its present form?

Yes

Are the interpretations and conclusions justified by the results?

Yes

Is the language acceptable?

Yes

Do you have any ethical concerns with this paper?

No

Have you any concerns about statistical analyses in this paper?

No

Recommendation?

Accept with minor revision (please list in comments)

Comments to the Author(s)

Krittika and Yadav show novel findings about the transgenerational effects of protein restriction (PR) on body size, body weight, water content, fat content, and wing length.

I have some concerns that should be addressed.

Major points

(1) Authors describe various changes in PR flies, but they are fragmented and very descriptive. Authors should clearly show their interpretation based on the results rather than only citing the contrary reports.

(2) In Fig.1, PR decreased body weight in generation 20. PR70 males decreased relative water content and increased relative fat content along with generation. In contrast, PR70 females increased relative water content and decreased relative fat content along with generation. Is there any physiological meaning in this sexual difference?

PR50 males decreased relative fat content along with generation just as PR70 females, and this is quite different with the result of PR70 males. Why does this difference occur between PR50 and PR70 males?

Minor points

(a) L218, "might to due to" is "might be due to" ??

(b) L224, "wing length" appears to be "pupal size".

(c) Fig.2 Bar style is not consistent between graph and legend.

Decision letter (RSOS-211325.R0)

Dear Dr Yadav

On behalf of the Editors, we are pleased to inform you that your Manuscript RSOS-211325 "Trans-generational effect of protein restricted diet on adult body and wing size of *Drosophila melanogaster*" has been accepted for publication in Royal Society Open Science subject to minor revision in accordance with the referees' reports. Please find the referees' comments along with any feedback from the Editors below my signature.

Please submit your revised manuscript and required files (see below) no later than 7 days from today's (ie 10-Dec-2021) date. Note: the ScholarOne system will 'lock' if submission of the revision is attempted 7 or more days after the deadline. If you do not think you will be able to meet this deadline please contact the editorial office immediately.

Please note article processing charges apply to papers accepted for publication in Royal Society Open Science (<https://royalsocietypublishing.org/rsos/charges>). Charges will also apply to papers transferred to the journal from other Royal Society Publishing journals, as well as papers

submitted as part of our collaboration with the Royal Society of Chemistry (<https://royalsocietypublishing.org/rsos/chemistry>). Fee waivers are available but must be requested when you submit your revision (<https://royalsocietypublishing.org/rsos/waivers>).

on behalf of Professor Laura Johnston (Associate Editor) and Steve Brown (Subject Editor)
openscience@royalsociety.org

Associate Editor Comments to Author (Professor Laura Johnston):

Associate Editor: 1

Comments to the Author:

Krittika and Yadav show novel findings about the transgenerational effects of protein restriction (PR) on body size, body weight, water content, fat content, and wing length.

I have some concerns that should be addressed.

Major points

(1) Authors describe various changes in PR flies, but they are fragmented and very descriptive. Authors should clearly show their interpretation based on the results rather than only citing the contrary reports.

(2) In Fig.1, PR decreased body weight in generation 20. PR70 males decreased relative water content and increased relative fat content along with generation. In contrast, PR70 females increased relative water content and decreased relative fat content along with generation. Is there any physiological meaning in this sexual difference?

PR50 males decreased relative fat content along with generation just as PR70 females, and this is quite different with the result of PR70 males. Why does this difference occur between PR50 and PR70 males?

Minor points

(a) L218, "might to due to" is "might be due to" ??

(b) L224, "wing length" appears to be "pupal size".

(c) Fig.2 Bar style is not consistent between graph and legend.

Reviewer comments to Author:

Reviewer: 1

Comments to the Author(s)

Krittika and Yadav show novel findings about the transgenerational effects of protein restriction (PR) on body size, body weight, water content, fat content, and wing length.

I have some concerns that should be addressed.

Major points

(1) Authors describe various changes in PR flies, but they are fragmented and very descriptive. Authors should clearly show their interpretation based on the results rather than only citing the contrary reports.

(2) In Fig.1, PR decreased body weight in generation 20. PR70 males decreased relative water content and increased relative fat content along with generation. In contrast, PR70 females

increased relative water content and decreased relative fat content along with generation. Is there any physiological meaning in this sexual difference?

PR50 males decreased relative fat content along with generation just as PR70 females, and this is quite different with the result of PR70 males. Why does this difference occur between PR50 and PR70 males?

Minor points

- (a) L218, "might to due to" is "might be due to" ??
- (b) L224, "wing length" appears to be "pupal size".
- (c) Fig.2 Bar style is not consistent between graph and legend.

===PREPARING YOUR MANUSCRIPT===

one version should clearly identify all the changes that have been made (for instance, in coloured highlight, in bold text, or tracked changes);

===PREPARING YOUR REVISION IN SCHOLARONE===

-- Ensure that your data access statement meets the requirements at <https://royalsociety.org/journals/authors/author-guidelines/#data>.

You should ensure that you cite the dataset in your reference list. If you have deposited data etc in the Dryad repository, please only include the 'For publication' link at this stage. You should remove the 'For review' link.

-- If you are requesting an article processing charge waiver, you must select the relevant waiver option (if requesting a discretionary waiver, the form should have been uploaded, see 'File upload' above).

-- If you have uploaded any electronic supplementary (ESM) files, please ensure you follow the guidance at <https://royalsociety.org/journals/authors/author-guidelines/#supplementary-material> to include a suitable title and informative caption. An example of appropriate titling and captioning may be found at https://figshare.com/articles/Table_S2_from_Is_there_a_trade-off_between_peak_performance_and_performance_breadth_across_temperatures_for_aerobic_scope_in_teleost_fishes_/3843624.

Author's Response to Decision Letter for (RSOS-211325.R0)

See Appendix A.

Decision letter (RSOS-211325.R1)

Dear Dr Yadav,

I am pleased to inform you that your manuscript entitled "Trans-generational effect of protein restricted diet on adult body and wing size of *Drosophila melanogaster*" is now accepted for publication in Royal Society Open Science.

Kind regards,
Royal Society Open Science Editorial Office
Royal Society Open Science

on behalf of Professor Laura Johnston (Associate Editor) and Steve Brown (Subject Editor)
openscience@royalsociety.org

Appendix A

Response to the Reviewer

Trans-generational effect of protein restricted diet on adult body and wing size of *Drosophila melanogaster*

Sudhakar Krittika and Pankaj Yadav*

Associate Editor Comments to Author (Professor Laura Johnston):

Associate Editor: 1

Comments to the Author:

Krittika and Yadav show novel findings about the transgenerational effects of protein restriction (PR) on body size, body weight, water content, fat content, and wing length.

Our response:

We thank the reviewer / Associate Editor for carefully reading our manuscript and giving us highly important and valuable corrections. We happily accepted these comments and have incorporated the changes *in toto*.

I have some concerns that should be addressed.

Major points

(1) Authors describe various changes in PR flies, but they are fragmented and very descriptive. Authors should clearly show their interpretation based on the results rather than only citing the contrary reports.

Our response:

We thank the reviewer /Associate Editor for pointing out the space for improvement and giving the insightful comment. As per the reviewer's suggestion, the results have now been explicitly mentioned and interpretation has been discussed. The lines indicating diverse contrary results have been marked with track changes.

Page 10, line 284-289,

Page 11, line 297-307, 312,

Page 12, line 334-335

(2) In Fig.1, PR decreased body weight in generation 20. PR70 males decreased relative water content and increased relative fat content along with generation. In contrast, PR70 females increased relative water content and decreased relative fat content along with generation. Is there any physiological meaning in this sexual difference?

Our response:

We thank the reviewer for pointing out and raising an important issue, in fact it is closer to the focus of the theme of the current manuscript. We completely agree with reviewer point/question. Yes, there might be some physiological background in this sexual difference. Firstly, the response of PR flies is the outcome of environmental factor (PR percent in diet in the tested generation), genetic factor (gross effect of past 20 generations) and the interaction factor of both environmental and genetic factors. In addition to this, there is chance of

phenotypic plasticity exhibited by these PR flies which is witnessed due to the long-term PR exposure and the probable reasons as indicated below:

Page 10, line 279-283

“Interestingly at gen 20, the dry weight of PR70 females is lower than that of AL, while its water content is higher and the fat content is similar to AL, thereby showing that the weight was probably contributed by the water content. But in case of males, they were similar to that of AL and their water and fat content are inversely proportional to each other.”

PR50 males decreased relative fat content along with generation just as PR70 females, and this is quite different with the result of PR70 males. Why does this difference occur between PR50 and PR70 males?

Our response:

We thank the reviewer for the valuable comment.

The difference between PR50 and PR70 males is probably due to the protein concentration difference. The results suggest that the PR50 males have not yet adapted to the storage of fat content across generations, while the PR70 males have. Interestingly, this is not the case in females, since protein is a highly important nutrient component for females, they might probably adapt quicker than the males, and stay along the same levels henceforth.

Since, our data is probably the one of the very few studies testing different concentrations of long-term PR diets, we still have unexplored the questions to be answered. But, with our current understanding of other tested traits across generations, we think this might be the possible reason for the witnessed effect.

Page 10, line 272-276

“There exists difference in the fat and water content between the males under PR50 and PR70, which could probably be due the difference in protein levels in the fly food. It can also suggest that adaptation to reduced protein level might be quicker and since, our data is one among the very few studies testing different concentrations of long-term PR diets, further in-depth studies are needed.”

Minor points

(a) L218, “might to due to” is “might be due to” ??

Our response:

We thank the reviewer for the comment. We highly regret the oversight and the correction has been made.

Page 8, line 223

“and might be due to the factors”

(b) L224, “wing length” appears to be “pupal size”.

Our response:

We regret the error and thank the reviewer for pointing out the same. The correction has been duly accepted and corrected.

Page 9, line 229

“ANOVA followed by Tukey’s HSD test on the pupal size showed”

(c) Fig.2 Bar style is not consistent between graph and legend.

Our response:

We regret the oversight and the corresponding corrections (the indication of the bars and legend has been consistently included in the figure).